

# Tuning path tracking controllers for autonomous cars using reinforcement learning

Ana Vilaça Carrasco and João Silva Sequeira

Lisbon University, Instituto Superior Técnico, Lisbon, Portugal

## ABSTRACT

This article proposes an adaptable path tracking control system, based on reinforcement learning (RL), for autonomous cars. A four-parameter controller shapes the behaviour of the vehicle to navigate lane changes and roundabouts. The tuning of the tracker uses an 'educated' Q-Learning algorithm to minimize the lateral and steering trajectory errors, this being a key contribution of this article. The CARLA (CAR Learning to Act) simulator was used both for training and testing. The results show the vehicle is able to adapt its behaviour to the different types of reference trajectories, navigating safely with low tracking errors. The use of a robot operating system (ROS) bridge between CARLA and the tracker (i) results in a realistic system, and (ii) simplifies the replacement of CARLA by a real vehicle, as in a hardware-in-the-loop system. Another contribution of this article is the framework for the dependability of the overall architecture based on stability results of non-smooth systems, presented at the end of this article.

# INTRODUCTION

Over the last decades, autonomous vehicles (AVs) have become a trendy research subject. The economics being developed around AVs is likely to have a significant societal impact, *e.g.,* reducing accidents and traffic congestion, optimizing energy use, and having an eco-friendly societal impact (the report by *Deichmann et al. (2023)* estimates revenues of $300–$400 billion by 2035, though also refering to the need for a change in mindset on the part of manufacturers), is a powerful motive for everyone working in this field.

The typical architecture of an AV, built around a guidance-navigation and control (GNC) structure, defines areas with multiple relevant research topics. From control topics, tightly connected to hardware and physical aspects, to planning and supervision, these have been addressed by multiple researches, see for instance the survey in *Pendleton et al. (2017)*. These include technical challenges, such as localization, trajectory following based on geometric principles, motion planning, communications, and vehicle cooperation, (*Omeiza et al., 2021*), but also societal concerns, namely explainability, (*Saha & De, 2022*), ethical, (*Hansson, Belin & Lundgren, 2021*), road infrastructure, *Manivasakan et al. (2021)*, and cyber security challenges, (*Algarni & Thayananthan, 2022*; *Kim et al., 2021*).

Corresponding author
João Silva Sequeira,
joao.silva.sequeira@tecnico.ulisboa.pt,
joao.s.sequeira@gmail.com

Regarding societal concerns, acceptance of AVs is still one of the biggest challenges faced by the industry. Results from surveys of the perception of US people were reported in _Rainie et al. (2022)_, showing a direct connection between acceptance, transparency/explainability, and the person's level of knowledge. Similarly, _Thomas et al. (2020)_ reported that people with higher education tend to be less concerned with liability issues. These results suggest that understandable, explainable systems will increase the acceptance of AVs by society.

AI techniques have been used in the design of the system's perception and guidance modules. _Devi et al. (2020)_ surveyed machine learning techniques applied to object detection, reporting that the best performance was obtained with convolutional neural networks (CNNs). _Grigorescu et al. (2020)_ reports an increased interest in using deep learning (DL) technologies for path planning and behaviour arbitration, with two of the most representative paradigms being planning based on imitation learning (IL) and deep reinforcement learning (DRL).

Following a reference trajectory, _i.e.,_ path following, is a key topic in generic AVs and has been extensively addressed in control theory and mobile robotics. Pure pursuit and Stanley methods conventional feedback controllers (_e.g._, linear quadratic regulators (LQRs), proportional integral derivative (PID) controllers) (_Yao et al., 2020_; _Sorniotti, Barber & De Pinto, 2017_), iterative learning control (ILC) _Yao et al. (2020)_, and model predictive control (MPC) (_Pendleton et al., 2017_) are examples that have been used for trajectory following. Less conventional control frameworks have also been proposed to tackle problems such as non-linearity, parameter uncertainties, and external disturbances, such as $H_\infty$, and sliding mode controllers (SMC) (_Yao et al., 2020_; _Sorniotti, Barber & De Pinto, 2017_).

Pure machine learning (ML) based solutions for vehicle control have also been proposed in the literature. _Kuutti et al. (2020)_ surveys a wide range of research in this area, reporting promising results from using DL methods for vehicle control, but also acknowledging significant room for improvement. A framework based on Q-Learning for longitudinal and lateral control is proposed in _Wang, Chan & Li (2018)_ and experiments involving the system while performing a lane change maneuver are presented. _Bojarski et al. (2016)_ proposes a supervised learning (SL) based end-to-end architectures with a complex NN that is able to replace a lane marking detection, path planning, and control pipeline. However, despite their popularity, these solutions tend to be computationally complex during training and the corresponding end-to-end architectures are associated with the ''black-box'' problem (_Kuutti et al., 2020_), lacking transparency and explainability. Simple fine-tuned feedback path tracking controllers have also been shown to provide good performance under a variety of conditions (_Sorniotti, Barber & De Pinto, 2017_). This suggests the use of a combination of these two architectures, by using supervision strategies to adjust/tune common controllers.

A variation of MPC, referred to as learning-based nonlinear model predictive control (LB-NMPC) is introduced in _Ostafew et al. (2015)_ to improve the path tracking in challenging off-road terrain. _Brunner et al. (2017)_ introduces a learning MPC variant to deal with repetitive tasks in the context of autonomous racing. The learning strategies in these variants are based on the application of optimization strategies to a space of parameters of processes modelling disturbances. Path tracking control approaches based

on DL have been shown to outperform traditional methods (*Li, Zhang & Chen, 2019*; *Shan et al., 2020*). Combining traditional path tracking controllers (*e.g.*, pure pursuit and PID) with reinforcement learning (RL) modules is proposed in *Chen & Chan (2021)*, *Shan et al. (2020)*, *Chen et al. (2019)*, and reported to effectively improve the performance of the traditional controllers. Using RL algorithms to optimize the parameters of PID controllers has also been proposed (*Ahmed & Petrov, 2015*; *Kofinas & Dounis, 2018*; *Shi et al., 2018*). RL and deep neural networks are used in *Shipman & Coetzee (2019)* to tune a PI controller for a simulated system, without any formal analysis. An Actor–Critic architecture to tune PIDs for the control of wind turbines is proposed by *Sedighizadeh & Rezazadeh (2008)*. Residual policy RL to tune a PID controlled car suspension is described in *Hynes, Sapozhnikova & Dusparic (2020)*, where it is shown that this method is sensitive to the initial conditions, *i.e.*, it may fail to adjust poorly tuned PIDs.

A variety of maneuvers, including lane keeping, lane change, ramp merging and navigating an intersection have been used to validate architectures (*Farazi et al., 2021*). In *Koh & Cho (1994)*, a set of maneuvers, namely, proceeding in a straight line, changing lanes, turning quickly, and circular turning, are used to validate their path tracking method. The same set of maneuvers are used in the present article to validate its proposed architecture.

The present article describes a path tracking controller using an RL agent to perform offline parameter tuning. The RL agent, using a discretized tabular variation of the Q-Learning algorithm (*Sutton & Barto, 2018*), is trained to fine-tune the controller's gains while performing lane changes and roundabout maneuvers.

The CARLA environment is used to simulate a car driving scenario. CARLA is an open-source simulator which has been reported to be suitable for learning applications in car driving (see *Malik, Khan & El-Sayed, 2022*; *Sierra et al., 2021*). By combining model-based controllers with learning algorithms, the properties of traditional controller designs can be preserved and the result still benefit from the robustness and adaptability of the learning component. By carefully constraining the parameter space explored during the learning phase(s), one obtains architectures that are dependable.

The proposed architecture also includes a module that identifies when to perform each maneuver and changes the gains appropriately, and a safety watchdog module that controls the vehicle's velocity.

This article is organized in the following sequence. The implementation section details the low-level path following controller and the supervising RL agent. Simulation results are presented next. The following section presents a framework based on non-smooth systems to model a supervised system of the kind described in this article and shows how dependability can be preserved. The section on conclusions summarizes the findings and points to future work. Portions of the text were previously published as part of a preprint (*Vilaça Carrasco & Silva Sequeira, 2023*).

## IMPLEMENTATION

The proposed architecture is presented in Fig. 1. The simulator implements a regular vehicle with multiple sensors attached. The use of the ROS framework (*Quigley et al.,*

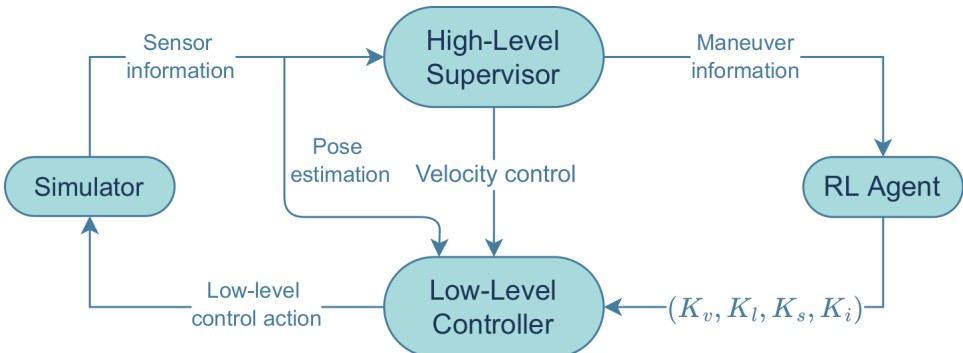

**Figure 1** **Figure with the full architecture used.** The main blocks of software are shown, together with the information exchanged among them.

*2009*) as a bridge between the vehicle and the overall control architecture allows a quick replacement of the simulator by a real vehicle. The low-level controller drives the vehicle through a predefined reference path by calculating and imposing values for the velocities and steering angles. The RL agent is trained to find the best set of gains for each maneuver. The maneuvers tested were lane changing (to the right) in a straight road and circulating in a roundabout.

The high-level supervisor monitors whether the linear velocity is within the imposed limits and whether the vehicle is required to perform one of the maneuvers. If so, it sends that information to the RL agent, which will then set the gains to the appropriate fine-tuned values for that maneuver. Those fine-tuned gains, denoted by $(K_v, K_l, K_s, K_i)$, are then sent to the low-level controller to calculate the steering angle, $\phi$, as well as the linear and angular velocities, $v$ and $w_s$–these three values define a low-level control action. If necessary, the value for $v$ is overridden by the high-level supervisor, to stay within the limits. The simulator also communicates directly with the low-level controller, sending an estimate of the vehicle's current pose, based solely on the vehicle's odometry.

## Low-level controller

The low-level control module controls the trajectory of the vehicle by adjusting the values of the steering angle $\phi$, linear velocity, $v$, and angular velocity, $\omega_s$, in real-time, with the goal of minimizing the error between the reference and the actual pose of the vehicle. The control laws, which are based on a nonholonomic vehicle model, are a function of the error between the reference and the actual pose, in the vehicle frame, $^b e$. The error in the world frame, $^w e$, is

$$^w e = [x_{ref} - x, y_{ref} - y, \theta_{ref} - \theta], \tag{1}$$

where $(x_{ref}, y_{ref}, \theta_{ref})$ is the reference pose and $(x, y, \theta)$ is the vehicle's current pose. The control laws to yield $v$, $\omega_s$ and $\phi$, written at discrete time $k$, are given by

$$v_k = K_v{}^b e_{x_k}, \quad \omega_{s_k} = K_s{}^b e_{\theta_k} + K_l b e_{y_k}, \quad \phi_k = K_i \phi_{k-1} + K_i h \omega_{s_k}, \tag{2}$$

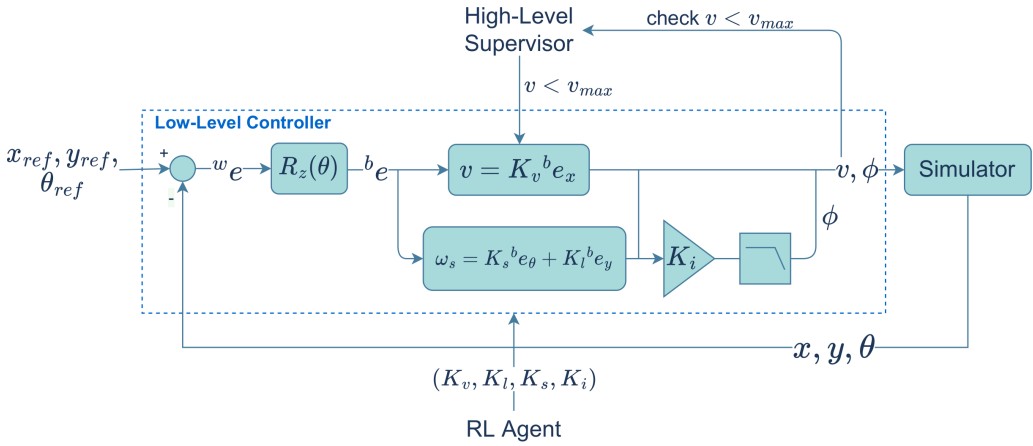

**Figure 2  Low level path tracking controller.**

where $h$ is a time step and ${}^b e$ is obtained from ${}^w e$ by means of a rotation matrix of a $\theta°$ rotation around the $Z$ axis. The last equation from Eq. (2) is a low-pass filter that removes unwanted fast changes in $\omega_s$.

The $v$, $\omega_s$ and $\phi$ are then converted into control actions and sent to the vehicle. The trajectory controller gains are the linear velocity gain, $K_v$, the steering gain, $K_s$, the linear gain, $K_l$, and the low-pass filter gain, $K_i$ (see Fig. 2).

The loop iterates until the destination is reached (*i.e.,* if the current position is close enough to the last position in the path), a collision is registered, or the simulation time ends.

## The simulator

CARLA (*Dosovitskiy et al., 2017)* is an open-source simulator designed for research on autonomous driving (*Chen et al., 2019*; *Samak, Samak & Kandhasamy, 2021*; *Shan et al., 2020*). It simulates urban realistic environments (in terms of rendering and physics). A ROS bridge allows direct communication with the simulated vehicle, through publishers and subscribers, and also provides a way to customize the vehicle setup. A "Tesla Model 3" vehicle was chosen, including speedometer, collision detector, and odometry sensors.

In this project, the simulator runs with a fixed time-step (the time span between two simulation frames) of 0.01 (simulation) seconds. Figure 3 illustrates, in a simple way, how the vehicle simulator transforms the linear velocity ($v$) and steering angle ($\phi$) provided by the low-level controller into messages that control the throttle, steer and brake values of the vehicle. The current pose of the vehicle is updated by subscribing to an odometry publisher provided by the simulator.

This simulator was run on a computer equipped with an Intel Core i5-7200U CPU (2.50 GHz ×4), and a NVIDIA Corporation GM108M [GeForce 940MX] GPU. The CARLA version used was 0.9.9.4, the CARLA ROS bridge version used was 0.9.8, with ROS Noetic in Ubuntu 20.04. More details about the simulator setup can be found

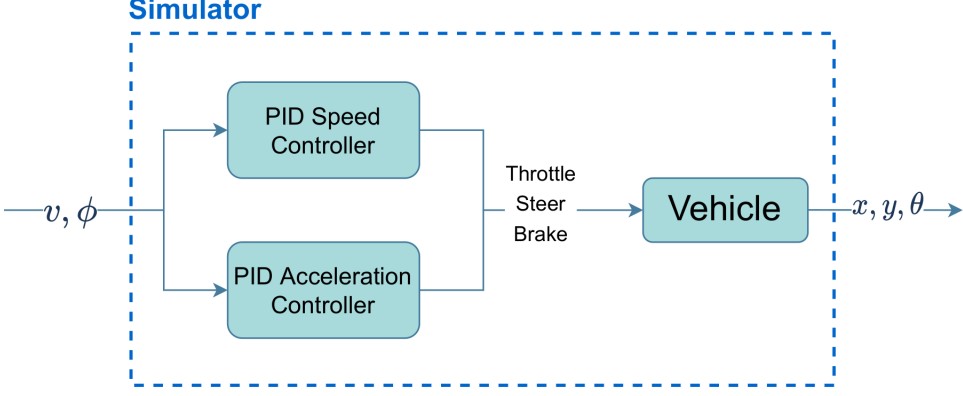

**Figure 3** Block diagram of the vehicle simulator.

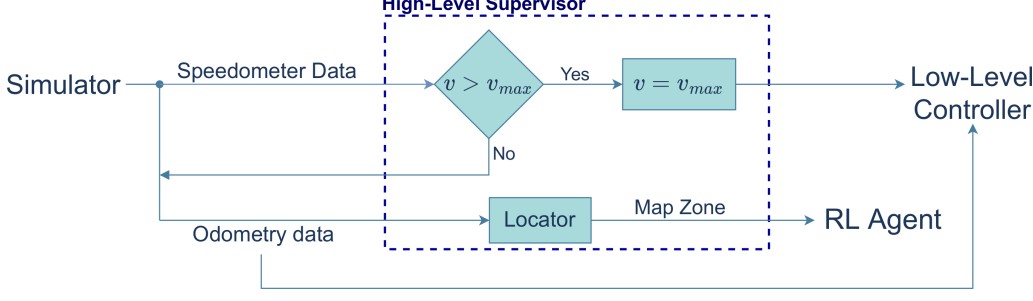

**Figure 4** Block diagram of the high-level supervisor controller.

on this project's GitHub and Zenodo (https://github.com/anavc97/RL-for-Autonomus-Vehicles)(https://doi.org/10.5281/zenodo.8078645).

## High-level supervisor

The high-level supervisor works as both an event manager and a safety module (see Fig. 4). It determines whether the vehicle needs to perform one of the two maneuvers based on which zone of the map the vehicle is currently in. It also enforces a speed limit, overriding, if necessary, the linear velocity calculated by the low-level controller.

In the experiments performed, the map was divided into zones, each of which associated with an event (see ahead in the article the definition of these zones). In the blue zone the vehicle performs a lane change and in the red zone, it navigates a roundabout. The reference path is shown as the black line.

## Reinforcement learning agent

The RL agent is responsible for tuning the trajectory controller gains using a variation of the Q-Learning algorithm. In the original Q-Learning, a discretized Q-Table takes an interval of gains and finds the values with which the vehicle presents the best performance. In the variation used in this work, referred to as educated Q-Learning, this interval of gains

is narrowed down to the most chosen values throughout the training. This facilitates the selection of the best gains by deliberately reducing the action space the algorithm has to explore. Performance evaluation is translated into the reward function of the algorithm. The RL environment is defined as follows.

**States:** An array with the average of the absolute values of the lateral and orientation errors, $S = [E_y, E_\theta]$. Each of the error values has low and high limits, $E_{LOW}$ and $E_{HIGH}$, and are discretized into 40 units.

**Actions:** Each action, $A = [a_0, a_1, a_2, a_3]$, is represented by an array. There are 81 different actions. The gains are adjusted by the action array using the following expressions,

$$K_v = K_v + h_0 a_0, \quad K_l = K_l + h_1 a_1,$$

$$K_s = K_s + h_2 a_2, \quad K_i = K_i + h_3 a_3. \tag{3}$$

The $a_0$, $a_1$, $a_2$ and $a_3$ can take the values 1, 0 or $-1$. The values $h_0$, $h_1$, $h_2$ and $h_3$ are positive constants that will either be ignored, subtracted or added to the previous value of the gain.

**Terminal condition:** Given the difficulty of reaching the state $S_0 = [0, 0]$, the adopted approach was to consider any state that would come closer to $S_0$ to be the terminal state. As a result, if the current state is closer to $S_0$ than the closest state recorded so far, then a terminal state was reached. To determine the distance from a state to the state $S_0$, $d(S, S_0)$, the algorithm uses a weighted Euclidean distance. Since the lateral error values, $E_y$, are generally 10 times greater than the orientation error values, $E_\theta$, the weight array used was $[1, 10]$:

$$d(S, S_0) = \sqrt{|E_y|^2 + 10 \times |E_\theta|^2}. \tag{4}$$

The sets of gains that produce the terminal states are referred to as the **terminal gains**. If, for the last 5 terminal sets of gains, a gain has a constant value, then that gain's range is locked into that value for the rest of the training –this defines the educated Q-Learning variation presented in this article.

Educated Q-Learning constrains the searchable space. The practical effect is the reduction of the volume of the searchable space, which does not compromise the convergence of the Q-learning. The unrestricted Q-Learning converges to a fixed point (or region). Following Schauder's fixed point theorem (see for instance *Bonsall (1962)*, Theorem 2.2), the restricted version also has a fixed point (region). Although Schauder's theorem requires the convexity of the reduced searchable space, (which, in general, will not be verified), the above idea continues to apply if we assume that the searchable space can be tessellated, *i.e.,* completely covered by a finite collection of convex regions. Therefore a trajectory in a convex cell either (i) converges to a fixed point inside, as a consequence of Schauder's theorem, or(ii) converges to the boundary of the convex cell from which it can be made to move into another cell (after which the convergence process re-starts in the current convex cell).

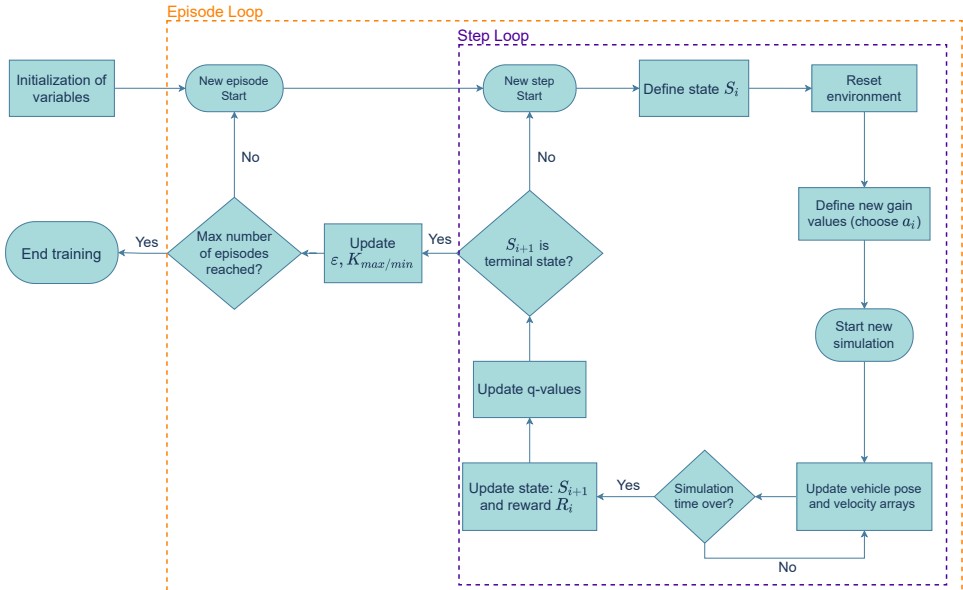

**Figure 5** Flowchart of the training algorithm.

In summary, constraining the searchable space in RL processes does not prevent them from reaching a solution. Even if the solution is potentially sub-optimal, we believe that it is close enough to optimal to produce satisfying results.

**Reward function:** The reward function chosen for this work is defined by the equation

$$R = \frac{1}{1 + d(S', S_0)} - \frac{1}{1 + d(S, S_0)}, \tag{5}$$

where $d(S', S_0)$ is the distance between the new state $S'$ and $S_0$ and $d(S, S_0)$ is the distance between the current state $S$ and $S_0$.

This function is based on the one used in *Kofinas & Dounis (2018)*. Also, if a collision is registered, the reward is decreased by a defined value.

**Training Algorithm:** The RL agent was trained to perform two different maneuvers: a lane changing maneuver in a straight road and driving in a roundabout. The algorithm used to train the RL agent is shown by the diagram in Fig. 5. The agent was trained over a certain number of episodes, each of which is divided by steps. In each step, a current state, $S_i$, is defined based on the last error average registered. Then, an action is taken and the new gains are defined, after which a new simulation starts, with the system's controller guiding the vehicle through the reference path. After the simulation stops, the new state, $S_{i+1}$, and the reward, $R_i$, are updated. With these values, the Q-Table is updated based on equation 6.8 in *Sutton & Barto (2018)* (p.131). If the new state, $S_{i+1}$ does not satisfy the terminal condition, this cycle repeats in a new step. Otherwise, the episode ends, $\varepsilon$ and the new gain range is updated (based on the educated Q-Learning variation) and a new episode starts.

**Table 1 Parameter values for each of the training environments, where $n$ is the number of episodes.**

| Variable | Lane change | Roundabout |
|---|---|---|
| Loop time | 5 | 30 |
| $\gamma$ | 0.9 | 0.9 |
| $E_{HIGH}$ (m) | [3 , 0.4] | [1 , 0.1] |
| $E_{LOW}$ (m) | [0 , 0] | [0 , 0] |
| $K_{min}(K_v, K_l, K_s, K_i)$ | [0.1, 1, 1, 0.7] | [1, 1, 1, 0.7] |
| $K_{max}(K_v, K_l, K_s, K_i)$ | [3, 21, 21, 0.98] | [5.8, 21, 21, 0.98] |
| $[h_0, h_1, h_2, h_3]$ | [0.58, 5, 5, 0.07] | [1.2, 5, 5, 0.07] |
| $\phi$ range (°) | ±30 | ±30 |
| $\varepsilon$ (start) | 1 | 1 |
| $\varepsilon$ decay | 1/(n/2) | 1/(n/2) |
| Step limit | 130 | 100 |

## SIMULATION RESULTS

The values chosen for the parameters, for each of the maneuvers, are shown in Table 1. The **Loop time** represents the time of each training step, in simulated seconds. $\gamma$ is the discount factor used for the Q-Learning equation used. $E_{HIGH/LOW}$ are the state limits (see the section on RL). $K_{max/min}$ define the range of gains explored. The values $[h_0, h_1, h_2, h_3]$ are the positive constants that define the action values (Eq. 3). The $\varepsilon$ **decay** and start value refer to the $\varepsilon$ greedy policy (*Sutton & Barto, 2018*) used in the Q-Learning algorithm. $\phi$ **range** is the default steering angle range used during training. **Step limit** refers to the maximum number of steps an episode can have, a condition that prevents unfeasible training times.

Using the algorithm in Fig. 5, the agent is trained to find the set of gains that minimize the error while the vehicle performs the two maneuvers. To choose the best set of gains, multiple training sessions were carried out, for each of the maneuvers, with different values for $\alpha$ (the learning rate). Figures 6 and 7 show the sum of rewards of each episode (learning curve).

Each training session had 30 episodes for the lane changing maneuver and 20 episodes for the roundabout navigation. The training times of these tests were, on average, 24 h and 33 h, respectively.

Both figures show a convergence of the learning curves, which implies the success of the algorithm.

To define the set of gains for each maneuver, we used the set of gains that was picked more times over all the different values of $\alpha$: **(3,21,21,0.7)** for the lane changing maneuver and **(3.4,21,1,0.84)** for the roundabout maneuver. These are the sets of gains used in the validation tests.

For the validation process, the system performs each of the maneuvers with the corresponding chosen sets of gains. Then, the average mean square error, $\overline{MSE}$, of the trajectory position is calculated by

$$\overline{MSE} = \frac{1}{N} \sum_{i=1}^{N} \frac{be_{x_i}^2 + be_{y_i}^2}{2}, \tag{6}$$

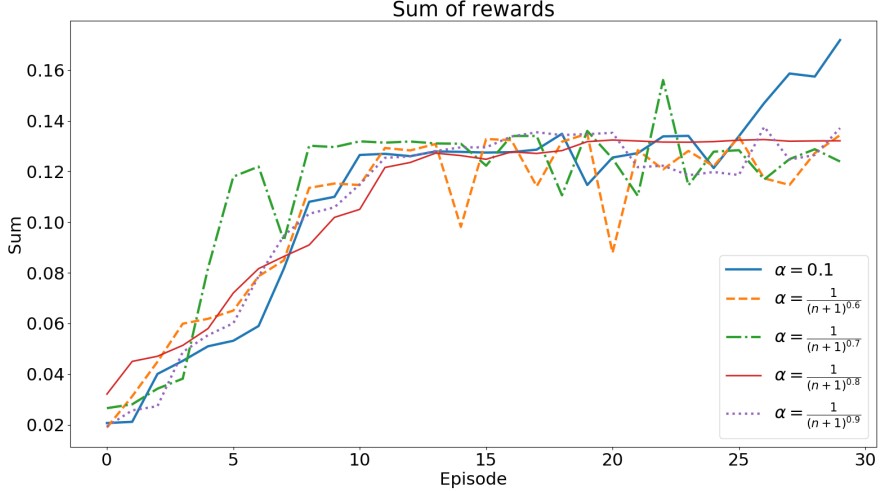

**Figure 6** **Accumulated rewards for lane changing maneuver.** Each curve shows the evolution for a specific value of the α parameter.

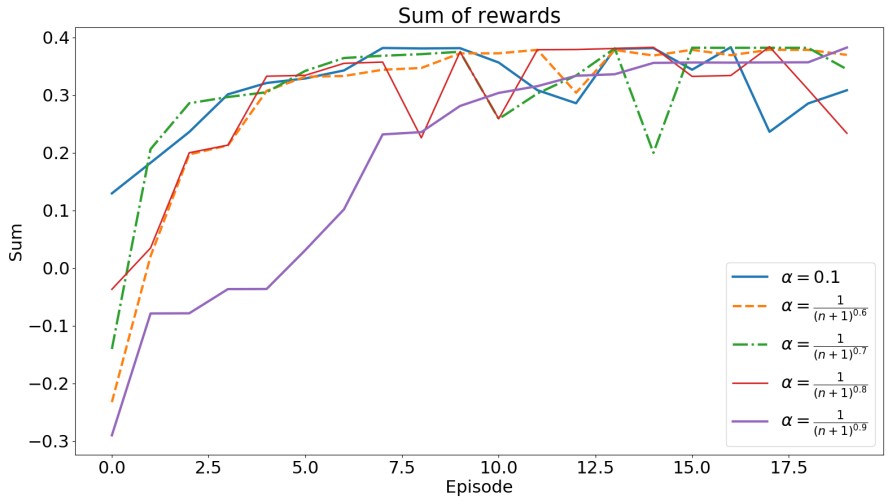

**Figure 7** **Accumulated reward for the roundabout maneuver.** Each curve shows the accumulated reward for a specific value of the α parameter.

where $N$ represents the number of data points registered. This process is repeated for different sets of gains spread over the range of values of the gain. The goal is to compare the performance of the chosen sets of gains with the performance of other sets of gains, while the system performs the maneuvers. Table 2 presents the average $\overline{\text{MSE}}$ for the lane changing and the roundabout maneuver. For each set of gains, the system performs the maneuver 10 times, and then the highest average MSE registered is selected.

By default, CARLA does not consider any noise in the odometry sensor. To analyse the robustness of the system, noisy odometry measurements were simulated. Position noise is

**Table 2** $\overline{\text{MSE}}$ (without noise) and $\overline{\text{MSE}}_\xi$ (with noise) for odometry measurements with different sets of gains, for the lane change and roundabout maneuvers.

| Lane change | | |
|---|---|---|
| **Gains** | $\overline{\text{MSE}}$ | $\overline{\text{MSE}}_\xi$ |
| $(0.1, 1, 6, 0.7)$ | 6.94 | 8.018 |
| $(0.68, 21, 21, 0.77)$ | 2.01 | 6.02 |
| $(1.26, 6, 11, 0.84)$ | 1.628 | 5.882 |
| $(3, 21, 16, 0.7)$ | 1.428 | 5.591 |
| **$(3, 21, 21, 0.7)$** | **1.359** | **5.589** |
| $(3, 21, 21, 0.98)$ | 1.399 | 5.637 |

| Roundabout navigation | | |
|---|---|---|
| **Gains** | $\overline{\text{MSE}}$ | $\overline{\text{MSE}}_\xi$ |
| $(2.2, 21, 1, 0.98)$ | 0.245 | 1.374 |
| $(2.2, 16, 21, 0.77)$ | 0.230 | 1.363 |
| $(3.4, 11, 21, 0.84)$ | 0.214 | 1.388 |
| **$(3.4, 21, 1, 0.84)$** | **0.208** | **1.347** |
| $(3.4, 21, 11, 0.77)$ | 0.216 | 1.385 |
| $(4.6, 6, 1, 0.84)$ | 0.265 | 1.429 |

**Notes.**
Min values are in bold.

obtained by drawing random samples from a normal (Gaussian) distribution with a mean of 0 and a standard deviation of 0.1 m. Orientation noise is obtained by drawing samples from the triangular distribution over the interval $[-0.088, 0.088]$ rad and centred at 0. The third column in Table 2 shows the $\overline{\text{MSE}}$ and $\overline{\text{MSE}}_\xi$ obtained under noisy conditions.

Figures 8 and 9 show the trajectory, linear velocity and angular velocity of the vehicle, without noise (blue) and with noise (red), and the reference path in orange, for the chosen gains. In the trajectory graph, the lengths of the trajectories with and without noise differ. This is because the duration of each validation test is fixed, and it takes longer for the system, with noisy odometry measurements, to take off. The delay in velocities, shown in the graphs below, corroborates this. For both maneuvers, these figures and Table 2 show small lateral errors and $\overline{\text{MSE}}$ values. A qualitative analysis of the values in Table 2 reveals that the gains chosen by the RL agent present the lowest $\overline{\text{MSE}}$, implying that the chosen gains are in the neighbourhood of the values that minimize the trajectory error. Furthermore, comparing $\overline{\text{MSE}}$ to $\overline{\text{MSE}}_\xi$, it is possible to verify the system's robustness to some noise in the odometry sensor measurements, in the sense that the chosen gains continue to produce the lowest values of $\overline{\text{MSE}}$.

The system was also tested while navigating in the environment illustrated in Fig. 10, following the reference path defined in red, which included both maneuvers and a sharp left turn. The chosen gains for the blue and red zones were, respectively, $(3, 21, 21, 0.7)$ and $(3.4, 21, 1, 0.84)$. For testing purposes, the speed limit imposed is 4 m/s. The results are presented in Fig. 11. It shows the trajectory performed by the system, without noise (blue) and with noise (red), and the reference path, in orange, which are, on average,

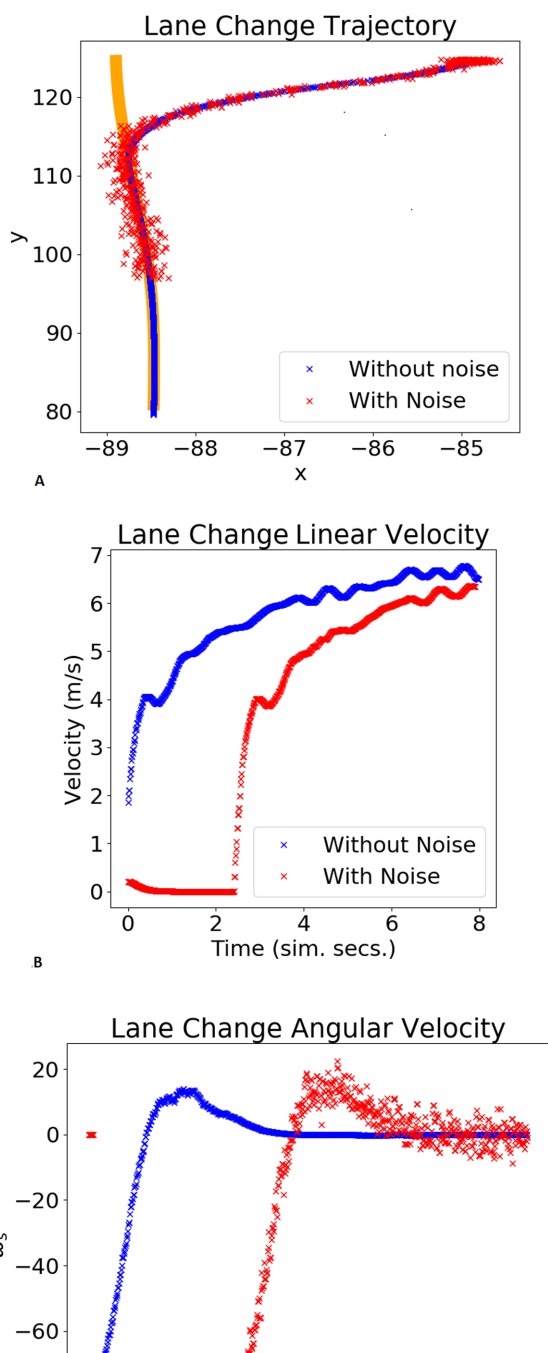

**Figure 8  Example of a lane change maneuver.** The plot shows the xy trajectory (in orange is the reference lane; in blue and red the trajectories obtained without/with noise).

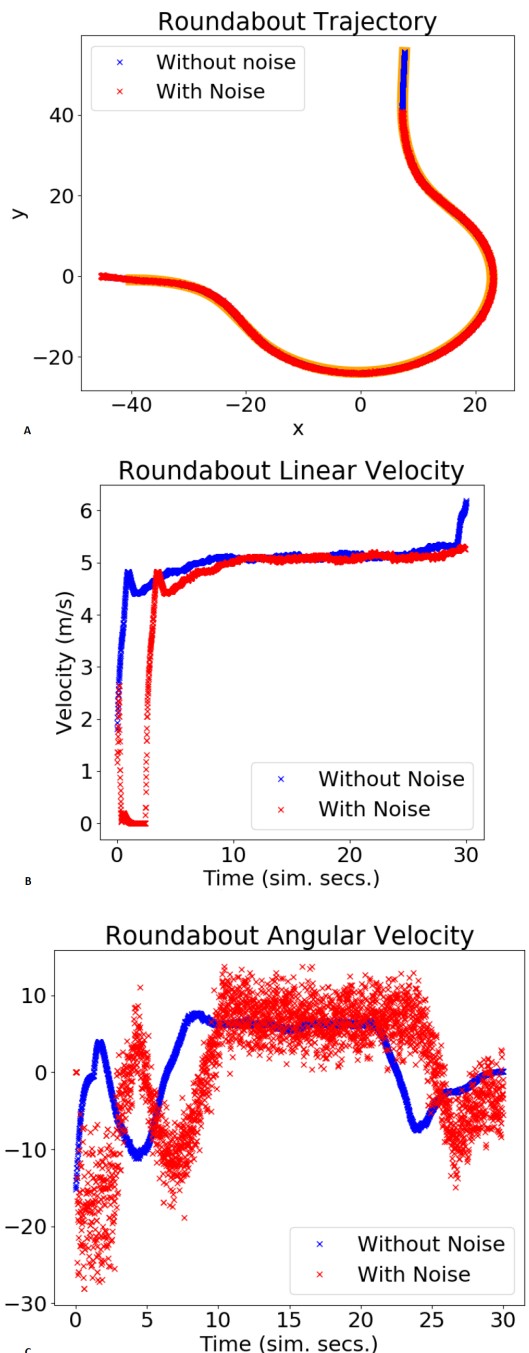

**Figure 9** **Example of trajectory for a roundabout maneuver.** The plot shows the xy trajectory (in orange is the reference lane; in blue and red the trajectories obtained without/with noise).

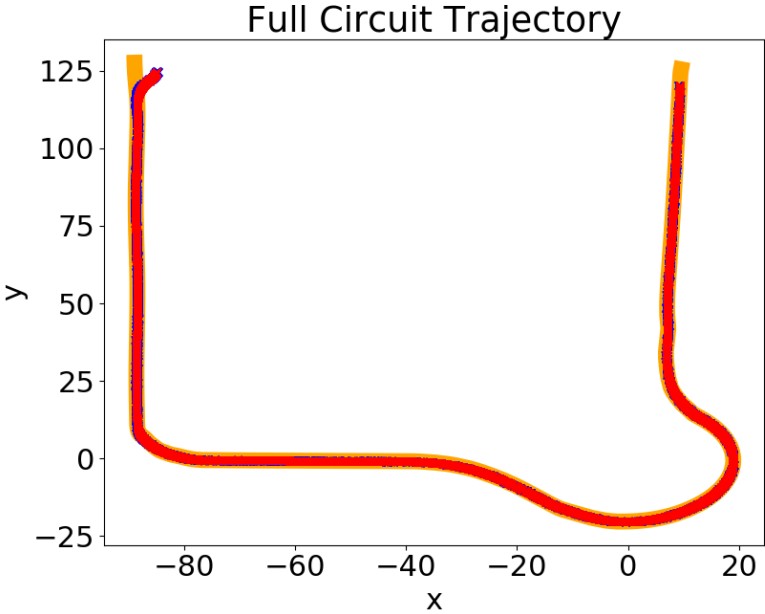

**Figure 10 Trajectory for a full circuit, including lane change and roundabouts.** Reference trajectory is shown in orange while the trajectory of the vehicle is shown in red.

superimposed: the system successfully follows the reference path, without any major errors or collisions.

Table 3 presents the values of $\overline{\text{MSE}}$ and $\overline{\text{MSE}}_\xi$ for different sets of gains, with the reference path defined in Fig. 10. As with the maneuvers, the results show that the chosen sets of gains produce the lowest $\overline{\text{MSE}}$ out of a wide range of gains, which suggests the proximity of the chosen gains to the optimal values of the gains.

## AN ARGUMENT ON DEPENDABILITY

This section aims at sketching a framework to research dependability properties in the RL-enabled autonomous car setting.

Dependability is tightly related to stability. Following *Avizienis et al. (2004)*, dependability means a consistent behaviour among different executions of the same task. Stability is a concept with many variations (*e.g.*, input–output, input-to-state) where some form of bounded behaviour is implicit. Furthermore, boundedness can be identified with the consistency that characterizes dependability, *i.e.,* in a dependable system state trajectories resulting from different executions will remain in a limited region of the state space.

The rationale behind the framework in this section is that dependability is ensured by using (i) a controller with a known topology and good stability and performance properties for a wide range of parameters, and (ii) an RL stage used to learn/select parameters for the controller adequate to each maneuver. Once stability is ensured for each individual scenario, it remains to derive conditions to ensure stability as the car switches between

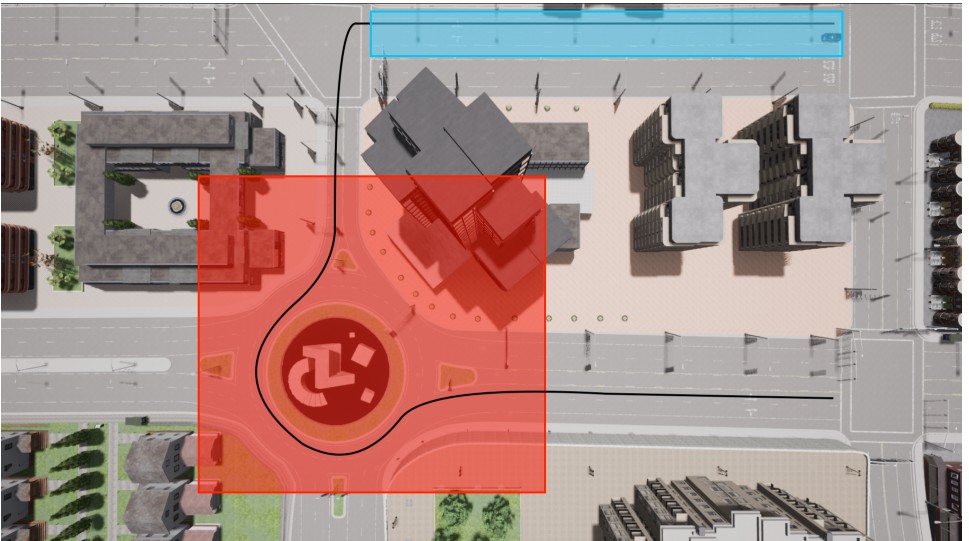

**Figure 11  Top view of the simulation environment with the reference trajectory.** The task includes a sharp left turn, lane change, and circulating on a roundabout.

scenarios.(As known from Control theory, switching between stable systems may lead to instability: see for instance (*Lin & Antsaklis, 2009*) in the framework of switched linear systems.) This is the concern addressed with this framework.

The GNC architecture, typical of a wide class of robotics systems, fully applies to the context of autonomous cars. The Control block accommodates multiple controllers tuned to specific driving conditions, *e.g.*, a trajectory tracker adapted to different maneuvers, such as overtaking maneuvers, changing between straight and twisty roads, or even changing between smooth and aggressive driving styles, and the Guidance block selects which of the controllers is used at every instant. Switching between controllers may be required in several situations, and hence the overall system is of a hybrid nature. Often, the switching mechanism will have the form of a finite state machine and the overall Control block can thus be described as an affine model,

$$C = C_0 + C_1 u_1 + C_2 u_2 + \ldots + C_n u_n \tag{7}$$

where $C_0, C_1, C_2, \ldots, C_n$ can be assumed state dependent smooth vector fields representing the output of each controller, and $u_1, u_2, \ldots, u_n$ stand for the switching control variables which are 0 whenever their respective controller is not active and $C_0$ is an affine term which may represent controller terms that must be always present (and not subject to any sudden change of structure).

Let $Q_1, Q_2, \ldots, Q_n$ be the accumulated reward trajectories for a set of $n$ controllers obtained during the RL training process, with each $Q_i$ corresponding to an individual maneuver. The value of an accumulated reward at the end of the training is an indicator of the quality of the policies found (*Sutton & Barto, 2018*, pp. 54–55). If the policies are allowed to run for a time long enough (so that they can reach their goals) the full system amounts to a sequence of individual/independent stable systems and is globally exponentially stable.

**Table 3** $\overline{\text{MSE}}$ and $\overline{\text{MSE}_{\xi}}$ for different sets of gains, for the full circuit.

| Gains | $\overline{\text{MSE}}$ | $\overline{\text{MSE}_{\xi}}$ |
| --- | --- | --- |
| $(1.84, 1, 1, 0.91)$,  $(2.2, 6, 11, 0.7)$ | 1.213 | 1.32 |
| **$(3, 21, 21, 0.7)$,  $(3.4, 21, 1, 0.84)$** | **0.181** | **0.363** |
| $(3, 21, 21, 0.98)$,  $(5.8, 16, 11, 0.84)$ | 0.185 | 0.673 |

**Notes.**
Min values are in bold.

However, in normal operation, each maneuver has a limited time/space to be completed and it may happen that the corresponding controller is not able to cope with it. Therefore, though each individual maneuver can be stable (and dependable) in unconstrained situations, arbitrary switching between maneuvers may rend the whole system unstable (hence losing the dependability property).

Using the converse Lyapunov theorem, this also means that there are Lyapunov functions $V_1, V_2, \ldots, V_n$, associated with each of the individual controllers, which, surely, have derivatives $DV_1 < 0, DV_2 < 0, \ldots, DV_n < 0$. Therefore, one can compose a candidate to Lyapunov function as

$$V = V_1 u_1 + V_2 u_2 + \ldots + V_n u_n \tag{8}$$

with $u_1, u_2, \ldots, u_n$ as defined above.

This technique has been reported in the literature when the $V_i$ are quadratic functions and the $C_i$ are polynomial vector fields (see for instance *Papachristodoulou & Prajna, 2002*, *Tan & Packard, 2004*). In this article we aim at a more general approach. The system formed by (i) the finite state machine structure used to switch between controllers, (ii) the controllers, and (iii) the car (assumed to be a regular kinematic structure such as the car-like robot) can be shown to be upper semicontinuous (USC). Writing (7) in the alternative set-valued map form as $C = C_0 \cup \left( \cup_{i=1}^{n} C_i u_n \right)$, following the definition of a USC set-valued map (see for instance, definition 1 in *Aubin & Cellina (1984)*, p. 41, or *Smirnov (2001)*, pp. 32–33), the overall system is USC as they have closed values and, by Proposition 2 in *Aubin & Cellina (1984)* this means that the corresponding graphs are closed. Hence, one is in the conditions required by the generalized Lyapunov theorem in *Aubin & Cellina (1984)* for asymptotic stability, namely

$$D_+ V(x) < -W(x). \tag{9}$$

If $W$ is a strictly positive monotonic decreasing function and $D_+ V(x)$ represents the contingent derivative of $V$ at $x$, $V$ is lower semi-continuous, then an equilibrium can be reached. This means that the overall system is dependable.

In general, in the car control context, the switching will occur at arbitrary instants, though a minimal separation between switching instants can be assumed without losing generality (as in a realistic situation a car will not switch arbitrarily quickly between behaviours in a repetitive way). Also, the switching will make $V$ have bounded discontinuities (at switching instants) and before each discontinuity will have a monotonic decreasing trend (as each behaviour is assumed asymptotically stable).

The $Q_i$ reference values are known *a priori* from the training phase. Monitoring the values obtained in real conditions and comparing them, in real time, with the training, yields a performance metric that can be used for control purposes, namely, defining thresholds to control the switching (*e.g.*, switch only if the currently observed $Q_i$ is close enough to the reference value recorded during training).

To illustrate the above ideas, consider the evolution of the accumulated reward function for different runs and parameters, shown in Fig. 6 for lane changes and in Fig. 7 for roundabouts. Each of these figures can be thought of as a set-valued map, $F(e)$, showing the evolution of the accumulated reward for each of the maneuvers. The convergence of each run ensures the boundedness of both the maps.

The images of $F(e)$ obtained at low episode values, $e$, indicate inefficient/incomplete executions. As the number of episodes evolves, the convergence of the RL finds efficient executions. At each episode, the images of $F(e)$ represent the intervals defined by the minimum and maximum of the accumulated reward. The $Q_i$ can be thus assumed to verify

$$\exists e \geq e_{\min} : Q_i \in F_i(e)$$

where $e_{\min}$ is a threshold defining the first episode from which learning is considered to be effective. In the case of the plots Figs. 6 and 7 a threshold of $e_{\min} \geq 10$ can be assumed as both plots show a plateau trend beyond this value. For the lane change plot $F_i(e_{\min}) \subseteq [0.08, 0.18]$ whereas for the roundabout one has $F_i(e_{\min}) \subseteq [0.19, 0.4]$.

The $F(e_{\min})$ obtained during training are implicitly referred to the 0 level, *i.e.*, the value of the accumulated reward at the beginning of the maneuver. Define the function representing the switching to the $i$th maneuver at time $t_i$ by

$$F_i(t) = \begin{cases} 0 & t < t_i \\ F_i(e_{min}) & t \geq t_i. \end{cases}$$

which indicates that the accumulated reward on a switching to the $i$th maneuver is represented by a bounded band starting at $t_i$ and being constant until $t_{i+1}$. During a normal, post-training, mission, with multiple transitions between $n$ maneuvers, the total accumulated reward can be written as a composition of shifted $F_i(\cdot)$ maps, as (assuming a sequence of $n$ maneuvers, which can also be assumed different without losing generality),

$$F(t) = F_1(t - t_1) + F_2(t - t_2) + \ldots + F_n(t - t_n)$$

where + is the Minkowski sum.

One can conceive multiple forms of estimating an enveloping function $W$. For example, consider the set $F_{ch} \equiv \text{ch}(\text{graph}(\{F(t_i)\}))$, with ch() standing for the convex hull operator, with $F(t) \subseteq F_{ch}$. The enveloping function $W$ can be constructed from the subdifferential of $F_{ch}$. By construction, $F_{ch}$ is compact, except if the switching between maneuvers occurs infinitely rapidly, in which case the accumulated reward grows without bound. An alternative form is to use a piecewise linear function $F_{bd}$ such that $F_{bd}(t) = F(t - t_i), \forall t_i$, that is enclose the accumulated reward $F$ by a piecewise linear function, from which the corresponding subdifferential can be easily computed.

## CONCLUSIONS

This article proposes an RL-based path tracking control system for a four-parameter architecture that minimizes the lateral and steering trajectory errors of the vehicle when performing lane changes and negotiating roundabouts. The tuning is done by a variant of the Q-Learning algorithm, here referred to as 'educated' Q-Learning, which reduces the action space during training, allowing a faster convergence to the final set of gains. An argument based on Schauder's fixed point theorem supporting the convergence of the proposed algorithm is presented.

The trajectories in Figs. 8, 9 and 11, as well as the velocities registered during these experiments demonstrate that the system does not engage in unsafe behaviour, such as collisions or excessive velocity. It also consistently follows the reference with little error. Although the proposed algorithm variant can lead to sub-optimal gains, the MSE values in Table 2 suggest that it can efficiently tune the gains to values that are in the neighbourhood of the values that minimize the trajectory errors (optimal gains). Additionally, the mean of the tracking errors registered for the lane changing, negotiating roundabouts, and a full circuit were, respectively, 0.076, 0.055 and 0.0166. These values are in line with others obtained for similar systems and testing conditions, namely (*Shan et al., 2020*; *Chen et al., 2019*).

Despite the popularity of NN based solutions for the control of an AV's path tracking, the proposed architecture offers higher explainability by providing control over the algorithm's state-action space, allowing the programmer to use traditional control design theory to ensure a stable behaviour. Additionally, the realistic simulation environment used is independent yet easily integrated, and the computational complexity of this system is lower than the NN alternatives, facilitating a smooth transition to real-world environments.

The framework for dependability developed in the previous section shows that the overall system has a stability property (which amounts to safe driving). Furthermore, given the rather general conditions imposed, the framework is applicable to other architectures that can provide a dependable performance indicator (such as the accumulated reward). Generic actor–critic architectures, *e.g.*, the supervised actor–critic in *Rosenstein & Barto (2004)*, are potential candidates to benefit from this framework.

Future avenues of research include (i) refining the dependability framework, namely to take into account the stochastic nature of the $Q_i$, and the educated Q-learning variation,(ii) exploring the use of dynamic reward functions, capable of representing different kinds of environments, *e.g.*, different road pavement conditions or different types such as urban/highway roads,(iii) further testing under noisy/uncertain conditions and design of behaviours to handle abnormal scenarios. The ultimate goal is the implementation in a real vehicle, where any unmodelled factors are likely to be a challenge.

The code developed for this work is available at GitHub and Zenodo (https://github.com/anavc97/RL-for-Autonomus-Vehicles, https://doi.org/10.5281/zenodo.8078645).

### Funding
This research was supported by FCT projects LARSyS LA/P/0083/2020 and UID-B/P/50009/2020. There was no additional external funding received for this study.

### Grant Disclosures
The following grant information was disclosed by the authors:
FCT projects: LARSyS LA/P/0083/2020 and UIDB/P/50009/2020.

### Competing Interests
The authors declare there are no competing interests.

### Author Contributions
- Ana Vilaça Carrasco conceived and designed the experiments, performed the experiments, analyzed the data, performed the computation work, prepared figures and/or tables, authored or reviewed drafts of the article, and approved the final draft.
- João Silva Sequeira conceived and designed the experiments, analyzed the data, authored or reviewed drafts of the article, worked out the technical arguments supporting the dependability of the whole architecture, and approved the final draft.

### Data Availability
The software is available at GitHub and Zenodo:

- https://github.com/anavc97/RL-for-Autonomus-Vehicles

- anavc97. (2023). anavc97/RL-for-Autonomus-Vehicles: v1.0.0 (05.2023). Zenodo. https://doi.org/10.5281/zenodo.8078645.

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
