# Peer review of "Tuning path tracking controllers for autonomous cars using reinforcement learning"

_PeerJ Computer Science, doi:10.7717/peerj-cs.1550_

## Round 0.1 · original submission · Major Revisions

Dear authors,

Your paper has been reviewed by two reviewers who asked for revisions of the paper. Please revise the paper according to comments by reviewers, mark all changes in new version of the paper and provide cover letter with replies to them point to point.

Reviewer 1 ·

Basic reporting

1. English writing is acceptable.
2. The paper does not have a proper literature review. The authors did not review the literature regarding all aspects of the investigated problem and the methodology used in the paper. The authors did not identify research gaps that their paper is trying to cover.

Experimental design

3. It is unclear from the abstract what the main contributions of this paper are.
4. Introduction is not written well. The authors should establish the main motives, aim, and research questions. In addition, the authors should highlight the main results, conclusions, and contributions of the paper. In addition, at the end of the introduction, the authors should provide a brief overview of the following sections (this is not mandatory but it is considered a standard in academic writing).

Validity of the findings

5. The authors should highlight what is novel in their methodology compared to the previous studies.
6. The paper does not have a proper discussion. The authors did not discuss how the results can be interpreted from the perspective of previous studies. Discussion should clearly and concisely explain the significance of the obtained results in order to demonstrate the actual contribution of the article to this field of research when compared with the existing and studied literature. In addition, the discussion should point out the theoretical and practical implications of the study.
7. Future research directions could be strengthened. They should be interesting to most of the Journal readership.

Additional comments

8. Acronyms/Abbreviations/Initialisms should be defined the first time they appear in each of three sections: the abstract; the main text; the first figure or table. For example, „CARLA“ and “ROS” are not defined in the abstract. Check the rest of the paper.
9. Some of the references are missing volume (and issue) numbers.

Cite this review as

Reviewer 2 ·

Basic reporting

All in all, the paper is good and well-written; however, there are also some issues that need to be addressed, namely:
1. The introduction becomes too fragmentary towards its ending - starting from line 56, it is actually written in a bullet-point style. In my opinion, the part of the introduction in question should be rewritten to make it more coherent.
2. Another issue with the same part of introduction is that interleaves the literature on the topic with the work done by the authors too much: on line 59, the authors start description of their own work, but then proceed with the discussion of the literature (Sutton and Barto, 2018; Koh and Cho, 1994; Wang et al, 2018) and get back to writing about their own work only in the very last sentence of the introduction. It would be helpful to divide these two lines of the narrative more clearly; furthermore, it would be helpful to conclude the introduction with a more thorough overview of the study. I would also suggest that the use of the CARLA simulator in the study should be mentioned in the introduction.
3. References should be put in brackets: for instance, it is 'a discretized tabular variation of the Q-Learning algorithm (Sutton and Barto, 2018)' rather than 'a discretized tabular variation of the Q-Learning algorithm Sutton and Barto (2018)'.
4. Some other issues with references:
- It is not clear what do Yao et al (2020) Pendleton et al. 30 (2017); Sorniotti et al. (2017)(Lines 29-30) actually reference: the original papers describing the methods? some survey articles on path tracking control methods? If it is the former, the references should be put right after the mention of the respective method rather than just be placed in the beginning of the list.
- Lines 34-35 appears to lack references for the methods mentioned there.
5. Some issues with grammar and formatting:
- Line 6: it seems that it can be removed, as it basically repeats line 5.
- Line 13 and elsewhere: no need to italicize ‘educated’
- Line 24: 'eco-friendly' rather than 'Eco-friendly'; 'impact on the world' rather than 'impact in the world'.
- Lines 26-27: ‘responsible for following’.
- Line 27: ‘ensuring that the system…’.
- Line 45: 'dependable by construction architectures' is awkward wording.
- Line 75: perhaps 'parameters' instead of 'gains'?
- 136 - 'Euclidean' rather than 'euclidean'.
6. Line 46: what 'Learning techniques' are meant here? classical ML algorithms?
7. A link to the Github repository should be included in the paper.

Experimental design

The experimental design of the study is quite solid and sound overall, yet I still think that it still should be improved in some respects, namely:
- While it is helpful indeed that the authors report the training times and some other details in lines 183-184, the paper is currently lacking the description of the setup, i.e., hardware and software apart from CARLA that was used in the study;
- The setup of the simulations involving CARLA should be described in greater detail: what version of CARLA was used? what map was used? what were other parameters of the simulation?

Validity of the findings

Due to the robust methodology of the study, it presents valid findings.

Additional comments

no comment

Cite this review as

---

## Round 0.2 · accepted · Accept

Dear authors,

Your paper has been accepted by one reviewer, while another not submit a review. According to the recommendation by one reviewer, your revised version of the paper, and your replies, my decision is to accept.

Reviewer 1 ·

Basic reporting

The authors have successfully addressed all issues from the previous review round.

Experimental design

The authors have successfully addressed all issues from the previous review round.

Validity of the findings

The authors have successfully addressed all issues from the previous review round.

Additional comments

The authors have successfully addressed all issues from the previous review round.

Cite this review as